# Is a COVID-19 Vaccine Likely to Make Things Worse?

**DOI:** 10.3390/vaccines8040761

**Published:** 2020-12-14

**Authors:** Stéphanie M. C. Abo, Stacey R. Smith?

**Affiliations:** 1Department of Applied Mathematics, The University of Waterloo, Waterloo, ON N2L 4A1, Canada; abostephanie@gmail.com; 2Department of Mathematics and Faculty of Medicine, The University of Ottawa, 150 Louis-Pasteur Pvt, Ottawa, ON K1N 6N5, Canada

**Keywords:** COVID-19, risk equations, vaccine, latin hypercube sampling, partial rank correlation coefficients

## Abstract

In order to limit the disease burden and economic costs associated with the COVID-19 pandemic, it is important to understand how effective and widely distributed a vaccine must be in order to have a beneficial impact on public health. To evaluate the potential effect of a vaccine, we developed risk equations for the daily risk of COVID-19 infection both currently and after a vaccine becomes available. Our risk equations account for the basic transmission probability of COVID-19 (β) and the lowered risk due to various protection options: physical distancing; face coverings such as masks, goggles, face shields or other medical equipment; handwashing; and vaccination. We found that the outcome depends significantly on the degree of vaccine uptake: if uptake is higher than 80%, then the daily risk can be cut by 50% or more. However, if less than 40% of people get vaccinated and other protection options are abandoned—as may well happen in the wake of a COVID-19 vaccine—then introducing even an excellent vaccine will produce a worse outcome than our current situation. It is thus critical that effective education strategies are employed in tandem with vaccine rollout.

## 1. Introduction

COVID-19 is a respiratory disease with flu-like symptoms that is the causative agent of a potentially fatal disease that has significant public-health concerns [1]. It was identified and gained traction in the city of Wuhan, in the Hubei Province of China at the end of 2019 [2] and was first identified in early December 2019 [3]. The most common symptoms at the onset of COVID-19 illness are fever, cough and fatigue; other symptoms include sputum production, headache, haemoptysis, diarrhea, dyspnoea and lymphopenia [4,5,6,7]. The period from the onset of COVID-19 symptoms to death, depending on the age of the patient and their immune status, ranges from 6 to 41 days, with a median of 14 days [6].

There were 425 laboratory-confirmed cases in Wuhan on 22 January 2020, leading to an initially estimated reproduction number of 2.2 (95% CI, 1.4 to 3.9) and a doubling time of 7.4 days [8]. By 16 February, the number of cases had climbed to 51,857 [1]. By 3 March, 90,870 cases of COVID-19 had been confirmed [9]. Multiple countries have since instituted temporary restrictions on travel, with an eye toward slowing the spread [10]. At the time of writing, there has been no vaccine released on the market, although two have been announced [11,12] and many are in development [13].

Once a vaccine is rolled out, not everyone will choose to get vaccinated or be able to, depending on immune status and cultural attitudes towards vaccines [14]. However, it is possible that when a vaccine becomes available, even if it not perfectly efficacious, much of the population may abandon other protection options, as has been theorised for HIV vaccines [15,16]. This may occur both for individuals who are vaccinated and those who are not.

Mathematical models can aid in predictions during the early stages of pandemics [17]. Early COVID-19 models have determined patterns of growth [18], the speed of spread [19], assessed existing outbreaks [20], made long-term predictions [21] and examined control strategies such as isolation of cases [22] or reduced social mixing [23].

Risk equations have been used to determine contributing morbidity and mortality factors for a variety of infections, such as heart disease [24], kidney failure [25], diabetes [26] and HIV [27]. Per-day risk equations to assess the potential effect of treatment have been used for vaginal microbicides as a potential HIV prophylaxis [28] and to assess the effect of potential Ebola vaccines [29]. Although the use of such equations in infectious disease modelling is relatively new, this kind of risk assessment has become an accepted component of clinical guidelines and recommendations in cardiovascular medicine, for example [30].

## 2. Methods

In order to evaluate the potential effect of vaccination, we develop risk equations for the per-day risk of COVID-19 infection both currently and after the introduction of a vaccine. When physical distancing, masks, handwashing or vaccination are employed, the probability that COVID-19 is transmitted will be reduced from the probability β to $\beta^{\prime }$ (where $\beta^{\prime }<\beta$). We have modelled single-exposure events, and thus the transmission probability of each event is significantly less than the transmission probability over the duration of the outbreak. If $\beta^{\prime }$ is the probability of transmission during a single exposure event with a given protection protocol (physical distancing, masks, handwashing, vaccination, any combination thereof, or no protection), then the probability of remaining uninfected during a single exposure event is $\left(1-\beta^{\prime }\right)$. The probability of remaining uninfected after *N* exposure events is thus ${\left(1-\beta^{\prime }\right)}^{N}$. Thus, the probability of COVID-19 infection for an individual is
$$\mathrm{Risk}=1-\;\prod_{i}^{\begin{array}{c} \mathrm{all}\;\mathrm{protection}\\ \mathrm{protocols} \end{array}}\;{\left[1-\beta^{\prime }\right]}^{p_{i}},$$
where $p_{i}$ is the proportion of times a given protection protocol is employed (including no protection).

The risk equations account for the basic transmission probability of COVID-19 (β) and the lowered risk due to various protection options: physical distancing ($e_{D}$), face coverings such as masks, goggles, face shields or other medical equipment ($e_{M}$), handwashing ($e_{H}$) and vaccination ($e_{V}$). We denote the probability of remaining uninfected with no protection by 1−β. Our risk equations also include the proportions of use (individually or in combination) of these protection protocols. We denote these proportions by $p_{i}$ if applied before vaccination and $q_{i}$ if applied after. The proportion of use of all protection protocols sums to 1 in each case.

The per-day risk before the introduction of a vaccine is thus given by
$$\begin{array}{cc} r_{1}=1- & {\left[1-\left(1-e_{D}\right)\beta \right]}^{p_{0}}{\left[1-\left(1-e_{M}\right)\beta \right]}^{p_{1}}{\left[1-\left(1-e_{H}\right)\beta \right]}^{p_{2}}\times \\ \; & {\left[1-\left(1-e_{D}\right)\left(1-e_{M}\right)\beta \right]}^{p_{3}}{\left[1-\left(1-e_{D}\right)\left(1-e_{H}\right)\beta \right]}^{p_{4}}\times \\ \; & {\left[1-\left(1-e_{M}\right)\left(1-e_{H}\right)\beta \right]}^{p_{5}}{\left[1-\left(1-e_{D}\right)\left(1-e_{M}\right)\left(1-e_{H}\right)\beta \right]}^{p_{6}}{\left[1-\beta \right]}^{1-\sum_{i=0}^{6}p_{i}}. \end{array}$$

Note that all the probabilities sum to 1 (including the possibility of no protection) so that these protection options, or combinations thereof, are exhaustive.

After the introduction of a vaccine, there is an additional protection option, as well as combined protection. The per-day risk after vaccination is given by
$$\begin{array}{cc} r_{2}=1- & {\left[1-\left(1-e_{D}\right)\beta \right]}^{q_{0}}{\left[1-\left(1-e_{M}\right)\beta \right]}^{q_{1}}{\left[1-\left(1-e_{H}\right)\beta \right]}^{q_{2}}{\left[1-\left(1-e_{D}\right)\left(1-e_{M}\right)\beta \right]}^{q_{3}}\times \\ \; & {\left[1-\left(1-e_{D}\right)\left(1-e_{H}\right)\beta \right]}^{q_{4}}{\left[1-\left(1-e_{M}\right)\left(1-e_{H}\right)\beta \right]}^{q_{5}}{\left[1-\left(1-e_{D}\right)\left(1-e_{M}\right)\left(1-e_{H}\right)\beta \right]}^{q_{6}}\times \\ \; & {\left[1-\left(1-e_{V}\right)\beta \right]}^{q_{7}}{\left[1-\left(1-e_{V}\right)\left(1-e_{D}\right)\beta \right]}^{q_{8}}{\left[1-\left(1-e_{V}\right)\left(1-e_{M}\right)\beta \right]}^{q_{9}}\times \\ \; & {\left[1-\left(1-e_{V}\right)\left(1-e_{H}\right)\beta \right]}^{q_{10}}{\left[1-\left(1-e_{V}\right)\left(1-e_{D}\right)\left(1-e_{M}\right)\beta \right]}^{q_{11}}\times \\ \; & {\left[1-\left(1-e_{V}\right)\left(1-e_{D}\right)\left(1-e_{H}\right)\beta \right]}^{q_{12}}{\left[1-\left(1-e_{V}\right)\left(1-e_{M}\right)\left(1-e_{H}\right)\beta \right]}^{q_{13}}{\left[1-\beta \right]}^{1-\sum_{i=0}^{13}q_{i}}. \end{array}$$

**Remark** **1.**
*We did not include the combination of all four protection options in $r_{2}$, on the assumption that a negligible fraction of people will go to this extreme once a vaccine is available.*


To determine the relative risk pre- and post-vaccination, we set the two risk equations equal to one another and calculated the risk threshold
$$\begin{array}{cc} r^{*}= & {\left[1-\left(1-e_{D}\right)\beta \right]}^{q_{0}-p_{0}}{\left[1-\left(1-e_{M}\right)\beta \right]}^{q_{1}-p_{1}}{\left[1-\left(1-e_{H}\right)\beta \right]}^{q_{2}-p_{2}}\times \\ \; & {\left[1-\left(1-e_{D}\right)\left(1-e_{M}\right)\beta \right]}^{q_{3}-p_{3}}{\left[1-\left(1-e_{D}\right)\left(1-e_{H}\right)\beta \right]}^{q_{4}-p_{4}}{\left[1-\left(1-e_{M}\right)\left(1-e_{H}\right)\beta \right]}^{q_{5}-p_{5}}\times \\ \; & {\left[1-\left(1-e_{D}\right)\left(1-e_{M}\right)\left(1-e_{H}\right)\beta \right]}^{q_{6}-p_{6}}{\left[1-\left(1-e_{V}\right)\beta \right]}^{q_{7}}{\left[1-\left(1-e_{V}\right)\left(1-e_{D}\right)\beta \right]}^{q_{8}}\times \\ \; & {\left[1-\left(1-e_{V}\right)\left(1-e_{M}\right)\beta \right]}^{q_{9}}{\left[1-\left(1-e_{V}\right)\left(1-e_{H}\right)\beta \right]}^{q_{10}}{\left[1-\left(1-e_{V}\right)\left(1-e_{D}\right)\left(1-e_{M}\right)\beta \right]}^{q_{11}}\times \\ \; & {\left[1-\left(1-e_{V}\right)\left(1-e_{D}\right)\left(1-e_{H}\right)\beta \right]}^{q_{12}}{\left[1-\left(1-e_{V}\right)\left(1-e_{M}\right)\left(1-e_{H}\right)\beta \right]}^{q_{13}}\times \\ \; & {\left[1-\beta \right]}^{\sum_{i=0}^{6}p_{i}-\sum_{i=0}^{13}q_{i}}. \end{array}$$

This quantity measures the change in the per-day risk after a vaccine is introduced. If $r^{*}>1$, then the vaccine is favourable (and hence the per-day risk will decrease after vaccination is introduced); if $r^{*}<1$, then the introduction of a vaccine increases the overall risk.

## 3. Results

### 3.1. Parameter Estimates

Matrajt and Leung examined the effectiveness of social distancing on COVID-19 and found that, for adults >60 years of age, when contacts for this group were reduced by 95% there was a moderate reduction of 21% fewer cases at the epidemic peak compared with baseline. In contrast, with >25% reduction in contact rates for the adult population, combined with 95% reduction in older adults, the number of hospitalizations and deaths could be reduced by >78% during the first 100 days of the epidemic. Similar numbers are found in Prem et al. [23], so we used the range $0.21\leq e_{D}\leq 0.78$.

Cotton masks lead to an approximately 20–40% reduction in virus intake compared to no mask. The N95 mask has the highest protective efficacy: approximately 80–90% reduction in virus intake. In contrast, cotton and surgical masks blocked more than 50% of the virus outward transmission, whereas the N95 masks showed approximately 90% protective efficacy [31]. The average filtration efficiency of single-layer fabrics and of layered combination was found to be 35% and 45%, respectively [32]. Both surgical masks and unvented N95 masks reduced the outward particle emission rates by 90% and 74% on average during speaking and coughing, respectively. Wearing a double-layer cotton T-shirt mask or a homemade cloth mask had no statistically significant effect on the particle-emission rate [33]. Hence, we used the range $0.2\leq e_{M}\leq 0.9$.

Proper hand hygiene has a 24–31% likelihood of decreasing the spread of transmissible disease [34], so we used the range $0.24\leq e_{H}\leq 0.31$.

The primary efficacy analysis of the Phase 3 Moderna study indicated a point estimate of vaccine efficacy of 94.5% (CI ) [12], while Pfizer/BioNTech reported a point estimate of 95% efficacy (CI ) [11]. To account for as-yet-unknown variations, we used a vaccine efficacy range of $0.85\leq e_{V}\leq 1$.

Since the exact transmission rate is unknown, we consider several scenarios, although we assume the rate is high, given the global nature of COVID-19. The ranges for each parameter are given in Table 1.

### 3.2. Before Vaccination

In order to assess the relative influence of each parameter on the outcomes, we performed a sensitivity analysis using Latin Hypercube Sampling and Monte Carlo simulations. Latin Hypercube Sampling is an efficient parameter-sampling process that uses the idea of Latin squares to avoid clustering and oversampling [35]. To determine the influence of each parameter, we calculated partial rank correlation coefficients (PRCCs). These values measure the relative sensitivity of the outcome to each parameter; if the PRCC >0, then the risk increases when that parameter is increased. Conversely, if the PRCC <0, then the risk descreases when that parameter is increased. Parameters are considered significant if |PRCC|>0.4.

Currently (i.e., before the introduction of a vaccine), the only protection options against COVID-19 are masks, distancing and handwashing. We considered two possibilities: (a) moderate transmissibility (75–80%) and (b) high transmissibility (85–90%). The PRCCs are shown in Figure 1.

In both scenarios of Figure 1, the most significant parameters are $p_{3}$ (the proportion of time a combination of physical distancing and masks are employed) and $p_{6}$ (when all three protection options are employed). It follows that both masks and physical distancing are important factors in managing COVID-19 disease in the absence of a vaccine.

In order to test the importance of our protection options, we stress-tested the outcome by systematically removing each of the options. See Figure 2. We simulated non-compliance with social distancing rules, refusal to wear face coverings (or shortage of masks) and poor hand hygiene. In each scenario, the most significant influence on the outcome is the proportion of use of the remaining options combined. When individuals abandon face masks, physical distance in combination with good hand hygiene ($p_{4}$) becomes critical for controlling the spread of the disease (Figure 2b). On the other hand, if proper hand washing is not enforced, physical distance and masks ($p_{3}$) become the most important factor in risk management of COVID-19 (Figure 2c). Interestingly, despite the differing efficacies, no single protection option emerged as superior to the others.

### 3.3. After Vaccination

The effectiveness of a vaccine depends upon the proportion of the disease burden that can be avoided by vaccination. As such, the efficacy of the vaccine, the percentage reduction in disease incidence in a vaccinated group and the vaccination rate are important factors in predicting the effectiveness of a potential vaccine. We tested the impact of vaccine uptake on reducing the risk of infection in the extreme case when vaccination is the only protection option employed, under the assumption that the degree of mitigation defiance seen in the pre-vaccination Western World will only expand once a vaccine becomes widely available.

Figure 3 shows the sensitivity of the daily risk of COVID-19 after vaccination to transmission rates for a variety of coverage levels. Figure 3a shows that, if the uptake rate varies widely ($0.25\leq q_{7}\leq 1$), vaccination will reduce the per-day risk, but the outcome also varies widely. In order to significantly decrease the daily risk of infection (i.e., individuals become less than 50% likely to get COVID-19 on any given day), Figure 3b shows that, broadly speaking, more than 80% of a population needs to be vaccinated (blue and green dots), depending on the transmissibility. Note that extremely low vaccine uptake (Figure 3b, purple dots) is worse than no vaccination (Figure 3a, red dots), since we assume that no other protection options are in place once a vaccine is available. It follows that even a vaccine with excellent efficacy, such as the Pfizer or Moderna vaccines, could make COVID-19 worse if vaccine uptake rates are too low.

In addition to the transmissibility, we also examined the sensitivity of the daily risk to overall vaccine uptake (Figure 4a) and the fraction of people using no protection (Figure 4b). Note that the latter is a dependent parameter, since all protection protocols sum to 1. Additionally, we examined the sensitivity of the daily risk to the vaccine efficacy. Since the efficacy is likely to be high, this was not a significant parameter (Figure 4c), except in the ideal case when update was perfect (Figure 4d), in which case higher efficacy can provide additional benefits.

Figure 5 illustrates three scenarios: currently (no vaccine); the vaccine being the sole protection option available used by some people (no protection is also a possibility); and the vaccine being the only protection option used by everyone. In the absence of a vaccine, the median risk probability $r_{1}$ is around 0.7. In the second scenario, the candidate vaccine is introduced, and people who initially used a protection option now entirely replace it with the vaccine, while those who have not used protection continue not to do so. The median post-vaccine risk probability $r_{2}$ decreases to 0.5 in this case. However, $r_{2}$ can vary widely ($0.01\leq r_{2}\leq 0.99$), and this volatility can be explained by the proportion of people using no protection. We can infer that the proportion of people using no protection has an important effect on the daily risk of infection and can significantly hinder vaccination efforts. In the third scenario, the ideal case of absolute vaccine uptake, the median per-day risk probability $r_{2}$ drops to 0.05, a net decrease from the initial risk $r_{1}$ of 0.7, with a corresponding decrease in volatility (0 $\leq r_{2}\leq$ 0.15). It follows that a high rate of vaccine uptake is necessary to significantly reduce the risk of COVID-19 infection.

### 3.4. Threshold Behaviour

The relative risk pre- and post-vaccination, $r^{*}$, measures the potential change in the per-day risk after the introduction of a vaccine. We recall that $r^{*}>1$ indicates a favourable vaccine, while $r^{*}<1$ indicates an increase in the overall risk after the introduction of a vaccine. Figure 6 shows that there is a threshold whereby the fraction of people who use the vaccine ($q_{7}$) is likely to increase the risk is if it sufficiently low (<40%) but likely to decrease the risk if it is sufficiently high (>50%), regardless of the protective behavior or lack thereof. It follows that the introduction of a COVID-19 vaccine may make the situation worse, unless at least half of the population has access to it. Figure 6 also shows that the more severe the disease (i.e, when transmission rates are high), the more benefit we derive from vaccination.

Figure 7 shows PRCCs for the parameters that affect $r^{*}$. If everyone abandons masks, physical distancing and washing hands, the outcome is very sensitive to vaccination uptake (Figure 7a). When people who initially used a protection option now combine it with the vaccine, while those who have not used protection continue not to do so, then supplementing the vaccine with additional protection options has a positive effect (Figure 7b). When social distancing is not possible, then vaccination alone is slightly more significant (Figure 7c), whereas if masks are not available, the most effective protection protocol is $q_{8}$, the combination of vaccination and distancing (Figure 7d). Finally, we note that the transmission rate (β) may in some cases have a (slight) positive influence on the risk threshold. This should be understood as follows: with higher transmission rates, the potential of any protection protocol to reduce the risk per day is greater, thus leading to larger values for $r^{*}$. Therefore, the positive correlation between transmission rates and $r^{*}$ is mediated by the stronger influence of protection protocols.

### 3.5. Discussion

Risk equations have the potential to predict the effect that introducing a vaccine can have on disease control. In our study, the parameters with the greatest impact on the COVID-19 pandemic are the transmission probability (β) and the fraction of people using the vaccine ($q_{7}$). Our results indicate that any effort to mitigate the COVID-19 crisis with a vaccine will have a significant positive effect on the overall risk if the majority of people receive the vaccine or if the vaccine is used to supplement existing protection measures.

The global community is working to reduce the spread of COVID-19 through aggressive measures such as nationwide lockdowns, restriction of social gatherings, contact tracing and isolation of infected individuals, as well as less stringent approaches such as physical distancing bylaws, mandatory use of masks and travel quarantine orders. This results in less contact between susceptible individuals and COVID-19 carriers. We have shown that the use of a vaccine in combination with these measures will reduce the per-day risk of infection so long as at least 50% of people receive it, with significant benefits if more than 80% people do. However, if there is too much vaccine defiance and a concomitant abandoning of other protection options, then we run the risk of a perverse outcome: the introduction of an excellent vaccine could nevertheless make the overall situation worse.

Our model has several limitations, which should be acknowledged. Initially, before a vaccine is introduced, we assumed a uniform use of all protection protocols available, including using no protection, which may not hold as populations become more aware of the disease and adjust their protective behaviours. We assumed the same efficacy for all types of face coverings, but performance varies between types of medical grade versus homemade masks [33]. After the vaccine is introduced, we assumed that other protection protocols are abandoned, which may not hold in practice, so this represents a worst-case scenario.

## 4. Conclusions

In summary, our results demonstrate the importance of vaccination in controlling the spread of COVID-19 disease. Highly efficacious vaccines, like those developed by Pfizer and Moderna, hold great hope of containing the pandemic. However, unless these vaccines are given to a sizable majority of people, vaccination is unable to fully replace existing protection measures. Until this goal is acheived, it is vital that public-health education about the importance of non-medical protection options remain in place.

## Figures and Tables

**Figure 1 vaccines-08-00761-f001:**
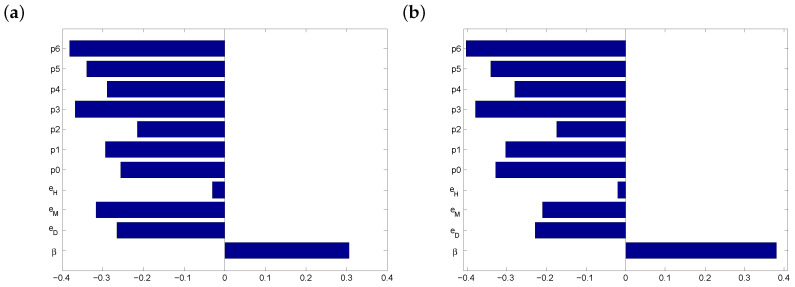
PRCCs before vaccine introduction for (**a**) 75–80% and (**b**) 85–90% levels of transmissibility. The outcome variable is the daily risk of infection before vaccination, $r_{1}$.

**Figure 2 vaccines-08-00761-f002:**
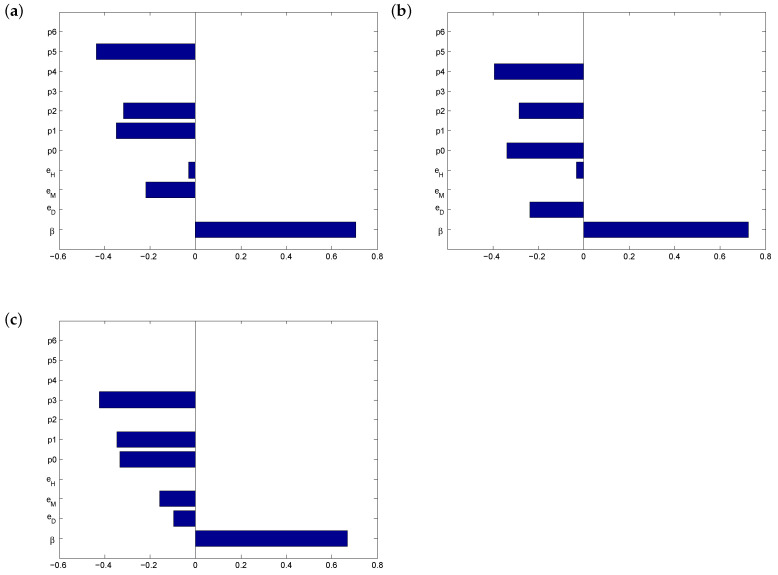
PRCCs for limiting cases: (**a**) non-compliance with physical distancing rules, (**b**) shortage of face masks and (**c**) no hand sanitizing. The outcome variable is the daily risk of infection before vaccination, $r_{1}$, and we consider high transmissibility (0.85≤β≤0.90).

**Figure 3 vaccines-08-00761-f003:**
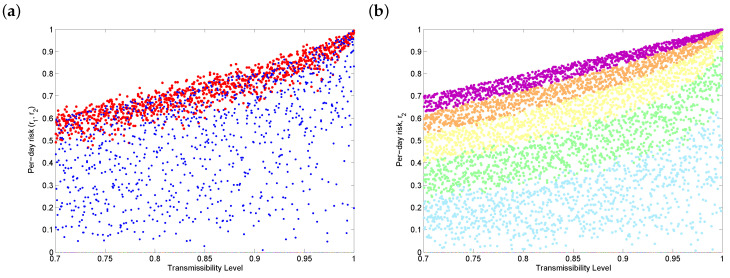
Sensitivity of the daily risk of COVID-19 to transmission rates when the only protection option available is the vaccine. (**a**) The red dots represent the pre-vaccination risk of infection, while the blue dots represent the post-vaccination daily risk. The fraction of people getting vaccinated, $q_{7}$, ranges from 0.25 to 1. (**b**) The outcome variable is the daily risk of infection after vaccination, $r_{2}$, and the fraction of people vaccinated, $q_{7}$, ranges from 0 to 0.2 (purple dots), from 0.2 to 0.4 (orange), from 0.4 to 0.6 (yellow), from 0.6 to 0.8 (green) and from 0.8 to 1 (light blue).

**Figure 4 vaccines-08-00761-f004:**
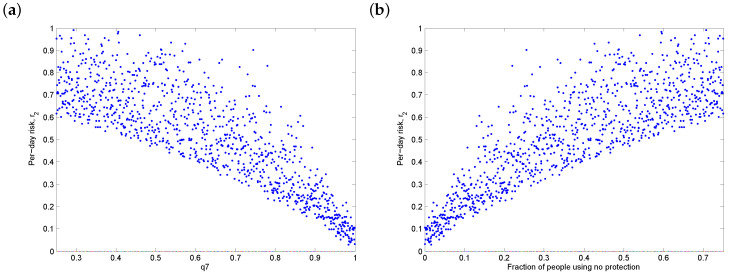
Sensitivity of the daily risk of COVID-19 to (**a**) vaccine uptake ($0.25\leq q_{7}\leq 1$), (**b**) the fraction of people using no protection, (**c**) vaccine efficacy and (**d**) the ideal scenario of perfect vaccine uptake ($q_{7}=1$). Note the reduced scale on the *y*-axis in the last figure. The transmission rate β is between 0.7 and 1.

**Figure 5 vaccines-08-00761-f005:**
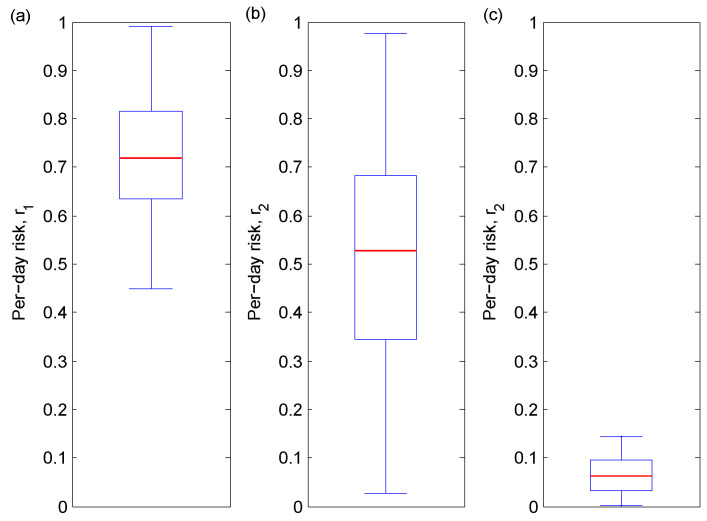
Boxplots of 1000 sampled values using the LHS ranges from Table 1, except with $q_{7}$ varying and other $q_{i}$ values set to zero. The horizontal red line indicates the median value. We present the daily risk of COVID-19 under three scenarios: (**a**) before the vaccine; (**b**) after the vaccine but with imperfect vaccine uptake ($0.25<q_{7}<1$); and (**c**) after the vaccine and with perfect vaccine uptake ($q_{7}=1$).

**Figure 6 vaccines-08-00761-f006:**
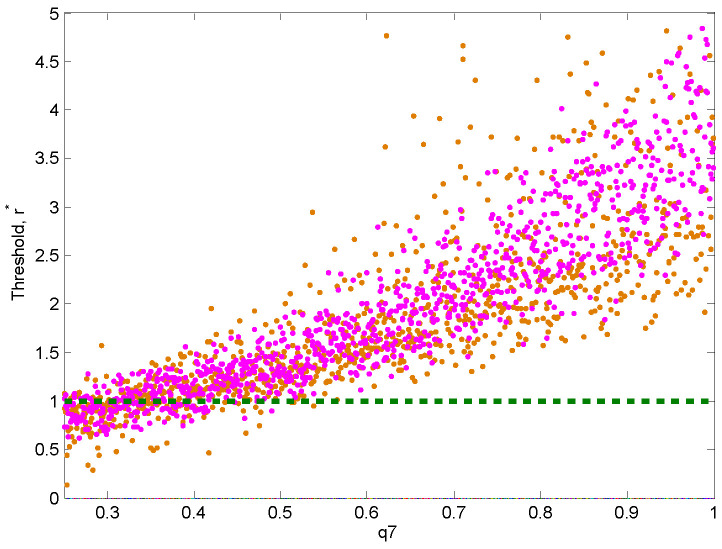
Sensitivity of the threshold level, $r^{*}$, to vaccine uptake, $q_{7}$. The transmission rate β ranges between 0.7 and 0.75 (brown dots) or between 0.85 and 0.90 (pink).

**Figure 7 vaccines-08-00761-f007:**
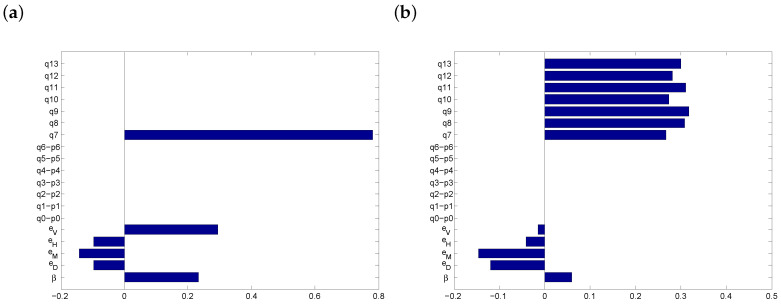
PRCCs after vaccine introduction. (**a**) Vaccination is the only option available; (**b**) all options and their combinations are in use; (**c**) when physical distancing is not possible; (**d**) when masks are not available. All values are as in Table 1, except that we constrained $0\leq q_{7}\leq 0.142$ in (**b**–**d**).

**Table 1 vaccines-08-00761-t001:** Parameter values before and after vaccine introduction. We assumed a uniform use of all available protection protocols (including no protection) before the vaccine. Thus, there are eight possible combinations of protection protocols: $p_{0}$ through $p_{6}$ and no protection ($1-p_{0}-p_{1}-p_{2}-p_{3}-p_{4}-p_{5}-p_{6}$). The upper bound of the range of values for each proportion is equal to 0.125. After the introduction of a vaccine, we considered both the extreme case where vaccination was the only available protection option ($0<q_{7}<1$) and the vaccine in combination with other protection options (with upper bound 0.142, including $q_{7}$).

Parameter	Definition	Range
β	Transmissibility	0.7–1
$e_{D}$	Physical distancing	0.21–0.78
$e_{M}$	Face coverings	0.2–0.9
$e_{H}$	Handwashing	0.24–0.31
$e_{V}$	Vaccine efficacy	0.85–1
$p_{0}$	Fraction of people practicing only physical distancing	0–0.125
$p_{1}$	Fraction of people only wearing face coverings	0–0.125
$p_{2}$	Fraction of people only practicing regular handwashing	0–0.125
$p_{3}$	Fraction of people practicing physical distancing and
	wearing face coverings	0–0.125
$p_{4}$	Fraction of people practicing physical distancing and
	regular handwashing	0–0.125
$p_{5}$	Fraction of people wearing face coverings with regular handwashing	0–0.125
$p_{6}$	Fraction of people practicing physical distancing with regular
	handwashing and wearing face coverings	0–0.125
$q_{0}$	Fraction of people only practicing physical distancing post-vaccination	0
$q_{1}$	Fraction of people only wearing face coverings post-vaccination	0
$q_{2}$	Fraction of people only practicing regular handwashing post-vaccination	0
$q_{3}$	Fraction of people practicing physical distancing and
	wearing face coverings post-vaccination	0
$q_{4}$	Fraction of people practicing physical distancing and
	regular handwashing post-vaccination	0
$q_{5}$	Fraction of people wearing face coverings with regular
	handwashing post-vaccination	0
$q_{6}$	Fraction of people practicing physical distancing with regular
	handwashing and wearing face coverings post-vaccination	0
$q_{7}$	Fraction of people using only the vaccine	0–1
$q_{8}$	Fraction of people using the vaccine and practicing physical distancing	0–0.142
$q_{9}$	Fraction of people using the vaccine and wearing face coverings	0–0.142
$q_{10}$	Fraction of people using the vaccine with handwashing	0–0.142
$q_{11}$	Fraction of people using the vaccine, practicing physical distancing
	and wearing face coverings	0–0.142
$q_{12}$	Fraction of people using the vaccine, practicing physical distancing
	with handwashing	0–0.142
$q_{13}$	Fraction of people using the vaccine, wearing face coverings
	with handwashing	0–0.142

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
