# Peer review of "Is a COVID-19 Vaccine Likely to Make Things Worse?"

_vaccines, 2020, doi:10.3390/vaccines8040761_

Round 1

Reviewer 1 Report

Review of vaccines 1022444
Is a COVID-19 vaccine likely to make things worse?

This paper investigates a critical question: whether the net effect of a vaccination program will be positive, given that the availability of a vaccine tends to decrease the use of other mitigation strategies. Clearly just showing a vaccine to be effective in reducing transmission, in isolation from other considerations, is not sufficient to justify its deployment. This is probably an issue not being adequately considered in public health planning. Done right, this would be a very important paper to publish quickly so that the results can be widely known before public health policies are locked in to place. However, there are some reasons why I feel that this paper is not yet ready.
The presentation of the method needs to be more clear. The authors use the words options, protocols, and types, but it is difficult to understand what each of these means. The first paragraph of Methods defines types to refer to the 8 possible combinations of the 3 things that one can do or not do. The word protocol is used to mean the same thing in the explanation of the risk equation, but the following sentence uses protocol to refer to the 3 things that can be done rather than the 8 combinations of those things. Just in case the reader is not already confused, the sentence “Our risk equations also include the proportions of use (individually or in combination) of these protection protocols” (my emphasis) almost guarantees confusion. This would all be very clear if the authors chose one word to always refer to the 3 independent things one can do and a different word to always refer to the 8 mutually exclusive combinations of the 3 things. I would use options for the 3 things and protocols for the 8 combinations. Then it would be very clear to say that the three independent options can be combined in various ways into eight mutually exclusive protocols numbered 0 to 7 (I would take the null protocol as #0 rather than have it be unnumbered, as at present) and being employed with probabilities p_i summing to 1.

I find it disappointing that the chosen parameter values are hypothetical rather than being based on those reported in the literature. Each of the 3 values e_H, e_M, and e_D is drawn randomly from the range [0,1/8], as though we have no reason to think that there is any difference between them. However, research has shown that mask usage can be expected to reduce transmission by a factor of much more than a mere eighth (at least three quarters, from what I have seen), while other research showing that most COVID-19 transmission is through the air suggests that handwashing is probably far less useful. Similarly, we now know that vaccine efficacy is likely to be much higher than what appears in the paper (of course it is not the authors’ fault that this information was unavailable when they wrote the draft!). I like the idea of using ranges to make up for our not knowing the exact values, but the ranges ought to be based on what we do know.

I don’t see why there is any asymmetry in the results of either Figure 1 or Figure 2. The parameters are the same, so how can there be any difference between the importances of the three options? Aside from the asymmetry, the results of Figure 2 should be obvious: if one option is unavailable, both the efficacy and the employment of each of the others should have increased importance compared to the case where the first option is also available.

Once vaccination comes into play, it isn’t clear to me what is meant by vaccine uptake. If q_7=1, that does indeed mean that everyone has taken the vaccine; however, wouldn’t uptake be complete as long as q_7+q_8+…+q_13=1, given that protocols 7 through 13 all involve vaccination? Perhaps the authors mean to make these other probabilities be 0 on the grounds that people who are vaccinated will not think that additional protections are necessary. Given the amount of mitigation defiance seen throughout the western world even without a vaccine, this certainly seems likely. The description of the results would make sense to me if these vaccine+mitigation protocols had been omitted. Some of the other results take perfect uptake to be ‘q_3=1’, which makes me think that some details got changed in a recent revision, with some notation updates being missed.
Overall, I would say this is an excellent paper in the making, but one that needs to be more carefully tailored to COVID-19 data to be useful for assessing COVID-19 risk specifically.

Author Response

See attached pdf

Reviewer 2 Report

This very interesting paper investigates the impact of vaccination on the risk of transmission of Covid-19, depending on the efficacy of the vaccine and vaccination rate, and assuming modification of protective behaviour.

The results are thought-provoking, especially the threshold behaviour for low efficacy vaccines, where vaccination increases the transmission risk because of behavioural effects. It is also interesting that the vaccination uptake has more impact on transmission rate than vaccine efficacy.

However, I wonder if the effects of the other protective measures, discussed at the beginning in the "Before vaccination" section (for example, the relative importance of distancing), result from the assumed efficacies of these interventions. It is not sufficient to cite references for these parameters, as a reader would surely prefer reading the values in the paper instead of consulting the literature. A table of all the assumed efficacies would significantly improve the interpretation of the results.

Some minor misprints:

  1. Page 2, in Methods (3rd line after the section title): "… handwashing or vaccination is employed "should be "… are employed"
  2. Page 3, line after line 83: "… we set the two risk equal "should be "… the two risks".
  3. Page 8, line 124: "Currently …" should be "Current …"

I find it curious that one author's surname finishes in a question mark (consistently in affiliation and references).
